# The Influence of Family History of Neurodegenerative Disease on Adolescent Concussion Outcomes

**DOI:** 10.3390/jcm10030528

**Published:** 2021-02-02

**Authors:** Colt A. Coffman, Adam T. Harrison, Jacob J. M. Kay, Jeffrey P. Holloway, Michael F. LaFountaine, Robert Davis Moore

**Affiliations:** 1Concussion Health and Neuroscience Lab, Arnold School of Public Health, University of South Carolina, Columbia, SC 29208, USA; harri735@email.sc.edu (A.T.H.); kay2@email.sc.edu (J.J.M.K.); 2Department of Pediatrics—School of Medicine, University of South Carolina, Columbia, SC 29209, USA; Jeffrey.Holloway@uscmed.sc.edu; 3Department of Physical Therapy, School of Health and Medical Sciences, Seton Hall University, South Orange, NJ 07110, USA; lafounmi@shu.edu; 4Departments of Medical Sciences and Neurology, Hackensack Meridian School of Medicine, Nutley, NJ 07110, USA

**Keywords:** concussion, adolescence, neurodegenerative disease, heart rate variability, mental health, cognition

## Abstract

Evidence suggests that factors associated with a family history of neurodegenerative disease (fhNDD) may influence outcomes following a concussion. However, the relevance of these findings in adolescent populations has not been fully explored. Therefore, the present study sought to evaluate the relationship between fhNDD and neurological outcomes following an adolescent concussion. Data from a local pediatric concussion clinic were used to compare adolescents with (*n* = 22) and without (*n* = 44) an fhNDD. Clinical symptom burden, emotional health, cardio-autonomic function, and cognitive performance were assessed at initial (~2 weeks) and follow-up (~5 weeks) post-injury evaluations. Cardio-autonomic function was assessed at rest and during isometric handgrip contraction (IHGC). Results indicated no significant group differences in emotional health or cognitive performance. Across evaluations, those with an fhNDD exhibited greater somatic symptom severity, alterations in HRV at rest, and early blunted cardio-autonomic reactivity during IHGC compared to those without an fhNDD. These findings suggest that positive fhNDD is negatively associated with clinical symptomology and cardio-autonomic functioning following an adolescent concussion. Further, these findings encourage clinicians to utilize a comprehensive neurological evaluation to monitor concussion recovery. Future studies should look into exploring the role of specific neurodegenerative processes and conditions on concussion outcomes in adolescents.

## 1. Introduction

A concussion is a form of mild traumatic brain injury (mTBI) induced by a direct blow to the head, neck, or body that is transmitted to cerebral tissues [1]. These biomechanical forces result in cascading alterations in neurometabolic function, which subsequently produce transient neurological impairment [2,3,4]. Post-concussive symptoms are often heuristically divided into three clinical domains: somatic (e.g., headache, dizziness, nausea), affective (e.g., depression, anxiety, irritability), and cognitive (e.g., attentional deficits, poor memory) [5,6]. Emerging research suggests that physiological deficits (e.g., cardio-autonomic dysfunction) present alongside the conventional concussion sequelae [7,8,9]. While post-concussive symptoms are expected to resolve within a few weeks, a subset of individuals (~15–20%) will remain symptomatic beyond the typical window of recovery [10,11]. Compared to adults, adolescents display an increased risk of both sustaining concussion and experiencing prolonged recovery thereafter [12,13]. These findings are particularly concerning as persisting deficits following concussion not only negatively affect the quality of life and academic achievement [14,15] but may leave adolescents susceptible to abnormal neurological development, which may present as hyperactivity or sustained cognitive impairment [16,17]. If adolescents are susceptible to future morbidity as a result of sustaining a concussion, researchers and clinicians need to identify premorbid factors that may be culpable in modulating neurological outcomes after injury.

Several factors associated with abnormal concussion recovery have been identified, including biological sex [18], history of concussion [19], and prior diagnoses of psychiatric or neurodevelopmental disorders [20,21]. In light of prominent research that has linked repetitive TBI to the onset of neurodegenerative disease (e.g., chronic traumatic encephalopathy, Alzheimer’s, Parkinson’s and dementia) in retired athletes [22,23], investigators have begun to examine how genetic precursors of neurodegenerative diseases might influence recovery trajectories following a concussion. Genetic polymorphisms associated with neurodegenerative diseases (i.e., APOE-ε4 allele, COMT Val/Met alleles) have been linked to both structural and functional neuronal abnormalities in healthy individuals [24,25,26]. Extensive research suggests individuals carrying these polymorphisms may be at risk for poorer outcomes following moderate/severe TBI [27,28]. It is postulated that particular gene variants may also predispose individuals to unfavorable neurogenic responses following concussion [29]. The APOE-ε4 allele has been shown to moderate post-concussion symptom reporting and neurocognitive performance in adult athletes and veterans across recovery milestones [30,31,32]. However, these findings have not been observed in studies following adolescent concussion [33]. While informative, current research focusing on APOE-ε4 allele-dependent gene expression has failed to integrate other heritable factors associated with neurodegenerative diseases.

An investigation of composite factors, such as an immediate family history of neurodegenerative disease (fhNDD), would allow researchers to acknowledge the influence of several modifiable risk factors. To date, only one study has investigated the influence of fhNDD on concussion outcomes [34]. However, this study exclusively examined college-aged males, thus, limiting the generalizability of its findings. Therefore, the present study sought to prospectively examine the influence of an immediate fhNDD (i.e., biological parent/grandparent) on adolescent concussion outcomes. In accordance with the most recent Consensus Statement on Concussion in Sport [1], we utilized a comprehensive assessment battery that assessed: symptom burden, emotional health, cardio-autonomic function, and cognitive performance following adolescent concussion at two remote time points after injury. We hypothesized that adolescents with a positive fhNDD would exhibit diminished cardio-autonomic regulation and poorer cognitive performance compared to adolescents without an fhNDD following concussion.

## 2. Experimental Section

### 2.1. Participants

This study is a retrospective analysis of data extracted from part of a larger study on the clinical evaluation of concussion. Four-hundred fifty adolescents, between the ages of 12–17 years old, suspected of having sustained a concussion recently, were evaluated at a local pediatric concussion clinic. Concussion diagnoses were confirmed by an attending physician (J.P.H.) in accordance with the guidelines established by the Consensus Statement on Concussion in Sport and the American Academy of Neurology [1,35]. Of the adolescents that were screened at the clinic for the presence of a concussion, those individuals that received a concussion diagnosis within 30 days of injury and returned for a follow-up post-acute evaluation were screened for eligibility in this research study (Figure 1). Participants with pre-existing neurological conditions such as seizure, psychiatric condition(s), developmental disorder(s), or those that were taking medications with known actions on neurological or cardiac function(s) were excluded from the present study.

A comprehensive demographic/health-information survey given to parents/legal guardians were used to identify those with (fhNDD+) and without (fhNDD−) a family history of neurodegenerative disease. Family history of neurodegenerative disease was defined as having either a biological parent or grandparent with a medical diagnosis of Alzheimer’s disease, Parkinson’s disease, or non-Alzheimer’s dementia. The fhNDD− and fhNDD+ groups were matched 2:1 on key demographic and injury information, including age, sex, body mass index (BMI), race/ethnicity, concussion history, time since injury, and cause of injury. After triage, the resultant dataset included twenty-two adolescents with fhNDD+ and forty-four adolescents without fhNDD− for analyses.

### 2.2. Procedure

Following diagnosis and initial evaluation, concussed adolescents were instructed to return in 2–3 weeks for a follow-up evaluation. Data were de-identified prior to data collection and analyses. All study procedures were approved by the Health Sciences South Carolina institutional ethics review board (reference #: Pro00075286). At each visit, participants were asked to complete a comprehensive neurological assessment battery consisting of a self-reported clinical symptoms checklist, psychological questionnaires, assessments of cardio-autonomic function, and post-injury cognitive performance tasks. Figure 2 provides a visual representation of the testing battery at each evaluation.

### 2.3. Measures

#### 2.3.1. Clinical Symptoms

The Rivermead post-concussion symptoms questionnaire (RPQ) was used to assess self-reported symptom burden. The RPQ is a 16-item survey used to rate the current severity of symptoms compared to premorbid status following concussion [36]. Beyond total symptom burden, the RPQ provides valuable information regarding specific symptom clusters, across: somatic (e.g., headache, nausea, dizziness or photophobia), emotional (e.g., irritable, depressed, frustrated), and cognitive (e.g., forgetfulness, poor concentration, taking longer to think) symptoms [37]. Higher scores indicate a greater symptom burden.

#### 2.3.2. Emotional Health

The depression subscale of Beck Youth Inventories—Second Edition (BYI-2) was used to evaluate emotional health following a concussion. The BYI-2 is a 20-item self-report questionnaire consisting of items that assess depressive traits such as sadness, pessimism, guilt, loss of pleasure, and fatigue in youth [38]. The BYI-2 has shown acceptable test-retest reliability (0.74–0.93) and convergent validity with other instruments used to assess depressive symptoms in youth [38]. T-scores were calculated from raw total scores using sex- and age-adjusted normative data. Higher raw and standardized scores indicate worse depressive symptoms.

#### 2.3.3. Cardio-Autonomic Function

Heart rate variability (HRV) was used to index cardio-autonomic function. HRV quantifies the temporal beat-to-beat variations in heart rate that arise through the intrinsic interplay between parasympathetic and sympathetic nervous systems [39]. HRV recordings were collected through an EmWave Pro Plus infrared pulse plethysmograph ear sensor (HeartMath, Boulder Creek, CA, USA) in a temperature and light-controlled environment. Participants maintained a self-paced breathing rate in a seated position for the duration of a 5-min recording. Additionally, during 1-min recordings, participants maintained a resonant breathing frequency to further investigate vagal activity and respiratory sinus arrhythmia during rest and physical exertion (isometric handgrip contraction; IHGC) [40,41]. By investigating individual changes from rest to physical exertion, HRV reactivity to acute stressors can be quantified and identify any self-regulatory dysfunction [42]. Data reduction and computation of time-domain and frequency-domain parameters were conducted with Kubios HRV Standard, version 3.0.2 (Biosignal Analysis and Medical Imaging Group, Kuopio, Finland) in accordance with recommendations by the Task for Force of the European Society of Cardiology and North American Society of Pacing and Electrophysiology [43]. HRV data were visually inspected and corrected for artifacts, and a 10% Hanning window was applied to the corrected data.

Time-domain parameters included the standard deviation of NN intervals (SDNN) and root mean square of successive NN interval differences (RMSSD) [44]. SDNN is known to reflect total cardiac variability, while RMSSD primarily reflects vagal tone [44]. Frequency-domain parameters included low-frequency (LF; 0.04–0.15 Hz) and high-frequency (HF; 0.15–0.4 Hz) band power components derived via fast Fourier transformations [45]. LF and HF band power (msec^2^) were quantified and expressed in natural logarithm transformed units. LF band components are known to reflect baroreflex activity and total cardiac variability, while HF band components reflect respiratory vagal tone [45]. Frequency-domain parameters have not been validated for ultra-short-term recordings (<5 min); thus, they were not calculated for one-minute recordings [46].

#### 2.3.4. Cognitive Performance

A modified CogState brain injury testing battery (CogState Ltd., Melbourne, Australia) was used to assess cognitive performance. The modified battery tests key domains of cognitive function, including working memory (one-back task; ONB), executive function (Groton maze learning test; GMLT), and visual memory recall (Groton maze delayed recall; GMR). Primary outcome measures included total errors (Groton maze learning and recall), reaction time (one back), and performance accuracy (one back). The selected tasks have shown acceptable validity and good reliability across various age-groups and clinical populations [47,48,49]. T-scores were calculated from raw test scores using age-adjusted normative data. Higher raw and standardized scores indicate poorer cognitive performance.

### 2.4. Data Analysis

A priori power analysis (G*Power 3.1) [50], with an alpha = 0.05 and power = 0.80, estimated a participant sample of 34 was sufficient to detect moderate effect sizes (*ηp*^2^ = 0.06). All statistical analyses were conducted using SPSS software version 27.0 (IBM Corporation, Armonk, NY, USA). To account for skewed distributions of RPQ and BYI-2 scores, a natural logarithm transformation was applied. HRV reactivity was quantified by calculating the change in HRV between 1-min assessments (∆HRV = IHGC HRV − resonant resting-state HRV). Independent samples *t*-tests were used for continuous data, and chi-squared tests were used for categorical data to compare group differences in demographic (age, sex, BMI, race/ethnicity) and injury characteristics (concussion history, time since injury, and cause of injury). Outcome measures were examined via a series of 2 (group: fhNDD+, fhNDD−) × 2 (time: initial evaluation, follow-up evaluation) repeated-measures analyses of variance (rmANOVA). Significant interactions were further decomposed via post hoc independent samples t-tests with Bonferroni correction for multiple comparisons. Levene’s tests were utilized to evaluate violations of equal variances and were corrected accordingly in the event of a violation. Partial eta squared (η_p_^2^) values were calculated to estimate the magnitude of significant differences (0.01 = small, 0.06 = medium, 0.14 = large). Pearson’s correlations were used to examine the associations between outcome measures in both fhNDD+ and fhNDD− patients. A priori level of statistical significance was set to *p* < 0.05.

## 3. Results

Participant demographic and injury characteristic information can be found in Table 1. Demographic characteristics did not differ between fhNDD+ and fhNDD− participants (*p* ≥ 0.17). In terms of injury characteristics, there were no differences between fhNDD+ and fhNDD− participants across concussion history, cause of injury, or time from injury to initial/follow-up evaluations (*p* ≥ 0.64). Of the fhNDD+ group, 17 adolescents had a family history of Alzheimer’s disease, and 6 adolescents had a family history of Parkinson’s disease. Analyses did not reveal any differences in concussion outcomes between those with a family history of Alzheimer’s and Parkinson’s disease (*p* ≥ 0.05).

### 3.1. Clinical Symptoms

Descriptive statistics for self-reported symptom burden can be found in Table 2. A significant main effect of group was revealed for the RPQ somatic subdomain (*F*_(1,64)_ = 4.503, *p* = 0.038, η_p_^2^ = 0.066; Figure 3A), whereby those with an fhNDD+ reported more somatic symptoms than those without an fhNDD−. Repeated measures ANOVAs did not reveal any other main effects of group or group x time interactions for overall or subdomain RPQ scores (*F*_(1,64)_ ≤ 3.927, *p* ≥ 0.052). However, significant main effects of time were observed for all RPQ measures (*F*_(1,64)_ ≥ 37.795, *p* < 0.001; Table 2). Irrespective of fhNDD, symptom burden decreased from initial evaluation to follow-up evaluation.

### 3.2. Emotional Health

Descriptive statistics for self-reported depressive symptoms can be found in Table 2. Repeated measures ANOVAs did not reveal any main effects of group or group × time interactions for Beck Youth Inventories—depression scale (BYI-D) raw scores or T-scores (*F*_(1,64)_ ≤ 1.261, *p* ≥ 0.266; Figure 3D,E). Irrespective of fhNDD, BYI-D raw scores and T-scores decreased from initial evaluation to follow-up evaluation (*F*_(1,64)_ ≥ 7.221, *p* < 0.01; Table 2).

### 3.3. Cardio-Autonomic Function

Descriptive statistics for HRV during the five-minute self-paced breathing assessment are displayed in Table 3. A significant main effect of group was observed for SDNN (*F*_(1,64)_ = 9.081, *p* = 0.004, η_p_^2^ = 0.124), RMSSD (*F*_(1,64)_ = 4.929, *p* = 0.03, η_p_^2^ = 0.072), LF power (*F*_(1,64)_ = 10.176, *p* = 0.002, η_p_^2^ = 0.137), and HF power (*F*_(1,64)_ = 8.961, *p* = 0.004, η_p_^2^ = 0.123). Group comparisons revealed that regardless of evaluation, the fhNDD+ group displayed significantly altered HRV compared to the fhNDD− group during a short-term self-paced breathing assessment (Figure 4). Repeated measures ANOVAs did not reveal any significant main effects of time or group x time interactions for HRV during self-paced breathing (*F*_(1,64)_ ≤ 1.733, *p* ≥ 0.193). Pearson’s correlations revealed that lower SDNN (r = −0.513, *p* = 0.015), RMSSD (r = −0.590, *p* = 0.004), and HF power (r = −0.505, *p* = 0.017) at rest were correlated with greater somatic symptoms in concussed adolescents with an fhNDD+, but not in fhNDD− individuals during follow-up evaluation.

Descriptive statistics for HRV during one-minute resonant breathing assessment can be found in Table 4. A significant main effect of group was revealed for SDNN (*F*_(1,64)_ = 9.598, *p* = 0.003, η_p_^2^ = 0.130) and RMSSD (*F*_(1,64)_ = 8.658, *p* = 0.005, η_p_^2^ = 0.119) during resonant resting-state assessment. Similarly, a significant main effect of group was revealed for SDNN (*F*_(1,64)_ = 9.019, *p* = 0.004, η_p_^2^ = 0.124) and RMSSD (*F*_(1,64)_ = 11.705, *p* = 0.001, η_p_^2^ = 0.155) during the physical exertion assessment. Regardless of timepoint, the fhNDD+ group had lower HRV at resonant resting-state (Figure 5A,B) and during IHGC (Figure 5C,D) compared to the fhNDD− group. No significant main effects of time or group x time interactions were revealed for HRV during either resonant breathing assessment (*F*_(1,64)_ ≤ 3.619, *p’s* ≥ 0.062). However, significant group x time interactions were revealed for ∆SDNN (*F*_(1,64)_ = 11.002, *p* = 0.002, η_p_^2^ = 0.147) and ∆RMSSD (*F*_(1,64)_ = 12.032, *p* = 0.001, η_p_^2^ = 0.158). Bonferroni corrected post hoc analyses revealed lesser SDNN (−2.00 ± 2.26 vs. −14.96 ± 2.75; *p* = 0.001) and RMSSD (1.72 ± 3.03 vs. −11.25 ± 3.09; *p* = 0.01) change in response to IHGC in the fhNDD+ group compared to the fhNDD− group at initial evaluation (Figure 5E,F). Main effects for group and time were not revealed for measures of HRV reactivity (*F*_(1,64)_ ≤ 0.504, *p* ≥ 0.48; Table 4). Symptoms were not correlated to HRV metrics during IHGC in concussed individuals.

### 3.4. Cognitive Performance

Descriptive statistics for CogState measures can be found in Table 5. Repeated measures ANOVAs did not reveal any main effects of group for raw CogState measures or T-scores (*F*_(1,64)_ ≤ 2.133, *p* ≥ 0.149). Main effects of time were observed for GMLT errors (*F*_(1,64)_ ≥ 22.225, *p* < 0.001), GMR errors T-score (*F*_(1,64)_ = 4.115, *p* = 0.047), ONB reaction time (*F*_(1,64)_ ≥ 13.447, *p* < 0.001), and ONB accuracy (*F*_(1,64)_ > 11.609, *p* < 0.001), whereby cognitive performance improved from initial evaluation to follow-up evaluation. No significant group x time interactions were observed for any cognitive measures (*F*_(1,64)_ ≤ 3.388, *p* ≥ 0.070; Figure 6).

## 4. Discussion

The aim of the present study was to evaluate the relationship between immediate fhNDD and adolescent concussion outcomes by assessing clinical symptoms, mental health, cardio-autonomic function, and post-injury cognitive performance. Our findings indicated that fhNDD+ adolescents report greater somatic symptom severity than fhNDD− adolescents following a concussion. With regard to cardio-autonomic function, fhNDD+ adolescents displayed diminished HRV at rest and during IHGC compared to fhNDD− across evaluations. Further, fhNDD+ adolescents exhibited blunted HRV reactivity compared to fhNDD− adolescents during the initial evaluation, as indicated by a lesser change in SDNN and RMSSD from rest to IHGC. However, an fhNDD did not influence mental health or cognition following an adolescent concussion.

Although clinical management of concussion commonly relies upon subjective symptom reporting, the understanding of factors that contribute to the experience of these symptoms is tentative and incomplete. The present findings suggest that a positive fhNDD may impact symptom reporting following a concussion, as fhNDD+ adolescents reported greater somatic symptom severity compared to their counterparts. Our results align with prior findings that demonstrate individuals with potential genetic factors associated with an fhNDD report a greater prevalence of posttraumatic headache and physical symptoms following concussion [30,51]. Interestingly, greater somatic symptom severity was associated with lower HRV at rest in fhNDD+ individuals during follow-up evaluation. This suggests that differences in HRV and post-injury somatic symptoms in fhNDD+ adolescents could be linked through a “psycho-somatic” origin and not from actual pathology.

Autonomic dysfunction is commonly experienced in individuals with neurodegenerative disease [52,53,54] and adolescents males with a family history of neurodegenerative disease [55]. In general, lower resting HRV is associated with worse health outcomes across neurological populations [56,57] and may be indicative of maladaptive self-regulation in response to environmental demands [58,59]. Research has shown cardio-autonomic impairment following concussion may manifest as sympathetic hyperarousal (i.e., decreased HRV) at rest [60,61] or as an atypical HRV response to physiological stressors [62,63,64]. Accordingly, we observed an attenuated HRV during both rest and IHGC in fhNDD+ adolescents compared to fhNDD− adolescents across evaluation time points. The Vagal Tank theory assumes a relative degree of vagal withdrawal (e.g., decrease in HRV) is needed to support metabolically demanding tasks, such as physical exertion [42]. Preliminary evidence suggests that individuals may demonstrate blunted cardio-autonomic reactivity following concussion [65,66]. Importantly, we observed that fhNDD+ adolescents show minimal to no change in HRV in response to IHGC compared to fhNDD− adolescents during the subacute phase of recovery (<30 days of injury). Together, these findings suggest that an fhNDD may partially account for the heterogeneity of cardio-autonomic dysfunction observed following an adolescent concussion. These results also support the use of HRV in clinical practice as a metric of concussion recovery.

Research suggests that reduced brain “reserve” may contribute to prolonged recovery trajectories following concussion [67,68]. Two concepts of “reserve”, neural and cognitive reserve, have been proposed by researchers to account for individual differences in concussion outcome despite similarities in insult and/or neurodegeneration [69,70]. Neural reserve, or the “passive” model of the reserve, refers to the ability of anatomical brain structures to withstand relative neuronal loss without functional consequence. Cognitive reserve, or the “active” model of the reserve, refers to the functional capacity of neural networks to compensate for neurodegeneration and/or neuronal loss. Diminished neural resilience in both structure and function have been exhibited in those with an fhNDD [71,72,73]. Accordingly, heritable factors associated with neurodegenerative diseases have been linked to impaired neurocognitive recovery following concussion [34,74,75]. Contrary to our hypotheses and prior evidence, fhNDD+ adolescents did not display greater neurocognitive deficits compared to fhNDD− adolescents following a concussion. However, current literature has primarily examined these subtle relations in older individuals. Advancing age is associated with a progressive loss of compensatory neural reserve [76,77]. Therefore, the younger age of our sample may explain why differences in cognitive performance were not observed between groups. Furthermore, the cognitive tasks selected may not be sensitive enough to capture group differences between fhNDD+ and fhNDD− adolescents. Future studies should incorporate higher-level cognitive tasks for a more in-depth investigation into the effect of an fhNDD following an adolescent concussion.

To date, this study is the first to prospectively examine the influence of an immediate fhNDD on concussion outcome among adolescents. Our results reflect studies in other neurological populations, which suggest that non-modifiable risk factors associated with neurodegenerative diseases may predispose individuals to adverse outcomes following neurological trauma [78,79]. Although it is plausible that a particular genotype (family history of Alzheimer’s disease or Parkinson’s disease) may contribute to the current findings, it is difficult to determine the role of genetic variability among genes that have previously been associated with neurodegenerative diseases as genotyping was not part of the data collection efforts. While these neurodegenerative diseases may share similar pathological characteristics, the genetic interactions are notably distinct and complex in heritability and onset [80,81]. This is an important point as many researchers have sought to examine particular gene variants in regard to concussion recovery. Future studies should explore polygenic profiles rather than a specific gene to elucidate the relationship between fhNDD and concussion outcome. From a clinical perspective, it is also more feasible to administer a simple questionnaire than conducting patient genotyping.

Further, the present findings underscore the importance of a comprehensive concussion assessment as recommend by the most recent Consensus Statement on Concussion in Sport, including cardio-autonomic assessment [1]. Even so, this study does not examine potential group differences in vestibular function, balance, anxiety, and sleep disturbances, which are commonly affected following an adolescent concussion. Future research should seek to further explore the effect of fhNDD on these domains of function. Although the two groups had comparable cognitive performance, fhNDD+ adolescents exhibited increased somatic symptoms and diminished cardio-autonomic function following a concussion. Somatic symptoms following concussion can affect a patient’s day-to-day functioning and quality of life. Additionally, even subtle deficits in cardio-autonomic function may contribute to exercise intolerance, postural hypotension, fatigue, and persistent posttraumatic headache [82,83,84]. Therefore, the current findings indicate that greater somatic symptom severity and cardio-autonomic dysfunction appear to be related in fhNDD+ adolescents through an undefined mechanism. More specifically, additional research is needed to explain why somatic symptoms were associated with resting-state HRV but not IHGC at the respective evaluations. Beyond postponing return to activity, these symptoms can negatively affect a child’s quality of life [85]. Therefore, the authors recommend that clinicians incorporate pre-evaluation screening questions into their practice to identify adolescents with an fhNDD, as they may be at greater risk to experience a more adverse outcome following concussion compared to others with no fhNDD.

Though the present study contributes novel findings to extant knowledge regarding concussion, it is not without its limitations. First, our sample size was not large enough to detect a small but potentially meaningful difference between groups. Therefore, we cannot determine if subtle group differences, perhaps in cognition, are present between fhNDD+ and fhNDD− adolescents. Additionally, we cannot determine to what effect concussive injury is responsible for the observed findings as pre-injury baseline measurements were not obtained. Furthermore, we cannot rule out differences in other components of mental health, such as anxiety, as mental health evaluation was limited to depressive symptoms. Lastly, the true fhNDD+ sample is likely underestimated. We examined adolescents; thus, it is reasonable to assume that parents or grandparents may be younger than the typical age (50–80 years old) in which many neurodegenerative diseases are diagnosed.

## 5. Conclusions

In summary, the present study is the first to examine the influence of an immediate fhNDD on adolescent concussion outcomes. Our findings provide evidence that concussed adolescents with an fhNDD may exhibit greater somatic symptomology and cardio-autonomic dysfunction relative to those without an fhNDD. Thus, our findings provide an impetus for clinicians to include screening for fhNDD and use of HRV in their clinical assessment to gain a greater understanding of pathophysiological differences between patients. HRV is feasible for use in clinical settings as it is non-invasive, cost-effective, and can be used under various conditions (e.g., rest, exercise, cognition). Furthermore, measuring HRV does not require the level of training or space of other psychophysiological measures and can be collected from simple ear clips or 3-lead ECGs. Together our findings suggest that screening for fhNDD and using HRV may be simple and effective ways for clinicians to identify patients who are likely to experience persisting symptoms as well as pathophysiology following a concussion. Doing so may help clinicians to preemptively change management strategies to maximize positive outcomes. However, more research is needed to determine if the same results are observed in adults and geriatric populations before broadly implementing HRV for patients across the lifespan.

## Figures and Tables

**Figure 1 jcm-10-00528-f001:**
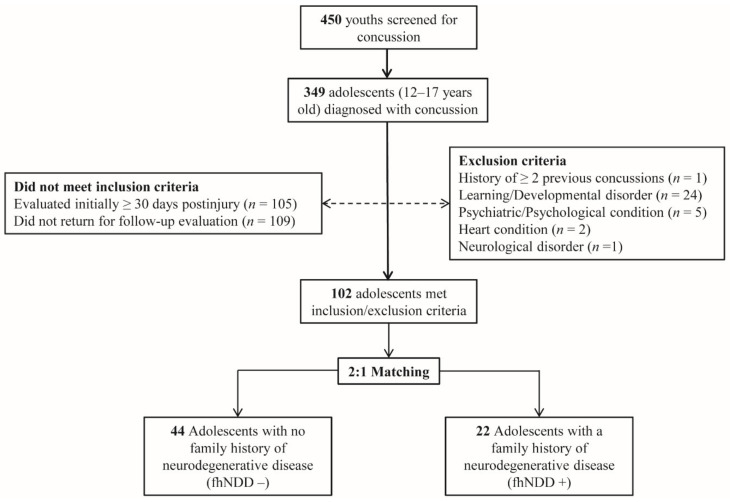
Flow diagram of sampled participants. Dotted line indicates excluded participants. Solid line indicates included participants.

**Figure 2 jcm-10-00528-f002:**
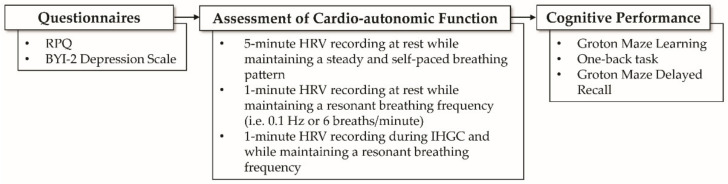
Visual representation of testing evaluation. Abbreviations: RPQ, Rivermead post-concussion symptoms questionnaire; BYI-2, Beck Youth Inventories—Second Edition; HRV, heart rate variability.

**Figure 3 jcm-10-00528-f003:**
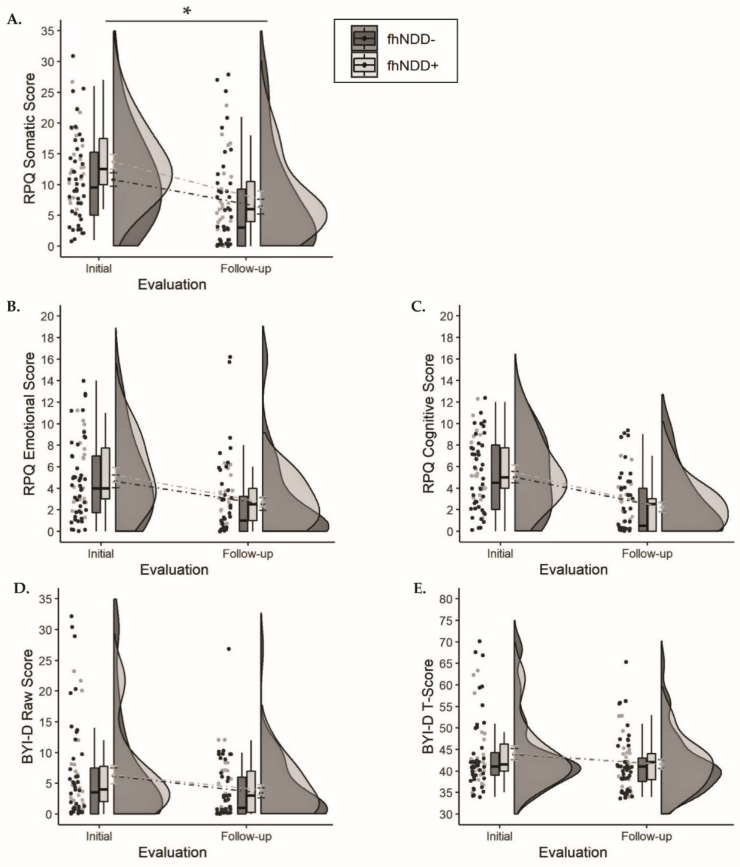
Group comparisons of symptom questionnaires between those with a family history of neurodegenerative disease (fhNDD+; light gray) and without a family history of neurodegenerative disease (fhNDD−; dark gray). (**A**) RPQ somatic score. (**B**) RPQ emotional score. (**C**) RPQ cognitive score. (**D**) BYI-D raw score. (**E**) BYI-D *t*-score. * *p* < 0.05.

**Figure 4 jcm-10-00528-f004:**
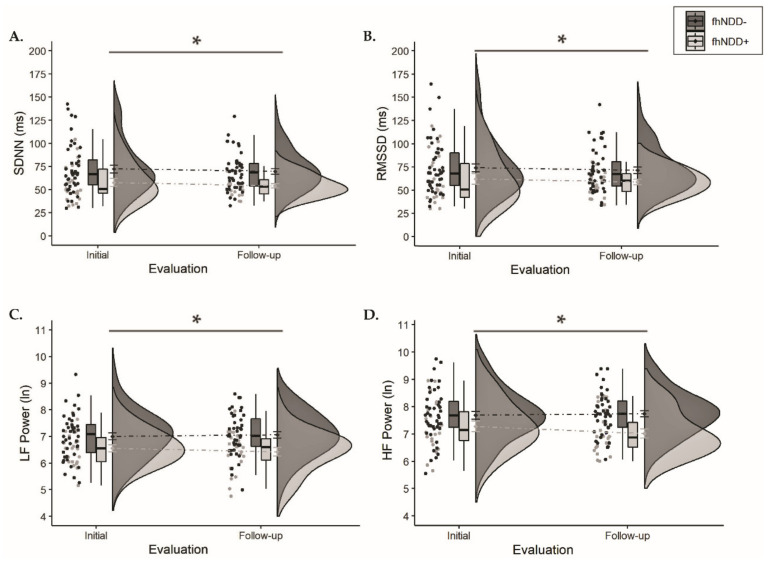
Group comparisons of HRV between those with a family history of neurodegenerative disease (fhNDD+; light gray) and without a family history of neurodegenerative disease (fhNDD−; dark gray) during self-paced breathing/resting-state assessment at each evaluation. (**A**) SDNN. (**B**) RMSSD. (**C**) LF power (natural logarithmic units). (**D**) HF power (natural logarithmic units). * *p* < 0.05.

**Figure 5 jcm-10-00528-f005:**
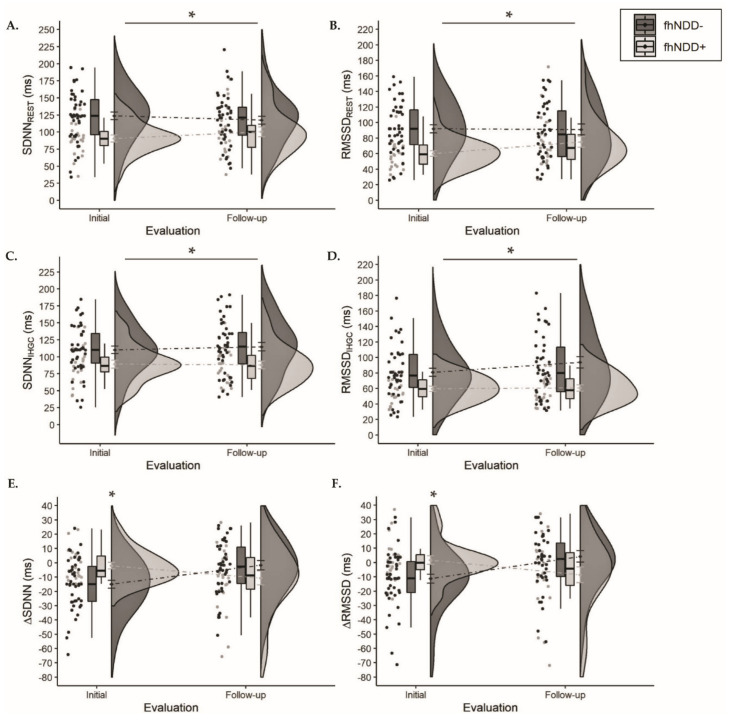
Group comparisons of HRV between those with a family history of neurodegenerative disease (fhNDD+; light gray) and without a family history of neurodegenerative disease (fhNDD−; dark gray) during resonant breathing assessments at each evaluation. (**A**) SDNN at rest. (**B**) RMSSD at rest. (**C**) SDNN during isometric handgrip contraction (IHGC). (**D**) RMSSD during isometric handgrip contraction (IHGC). (E) Change in SDNN from rest to IHGC. (**F**) Change in RMSSD from rest to IHGC.* *p* < 0.05.

**Figure 6 jcm-10-00528-f006:**
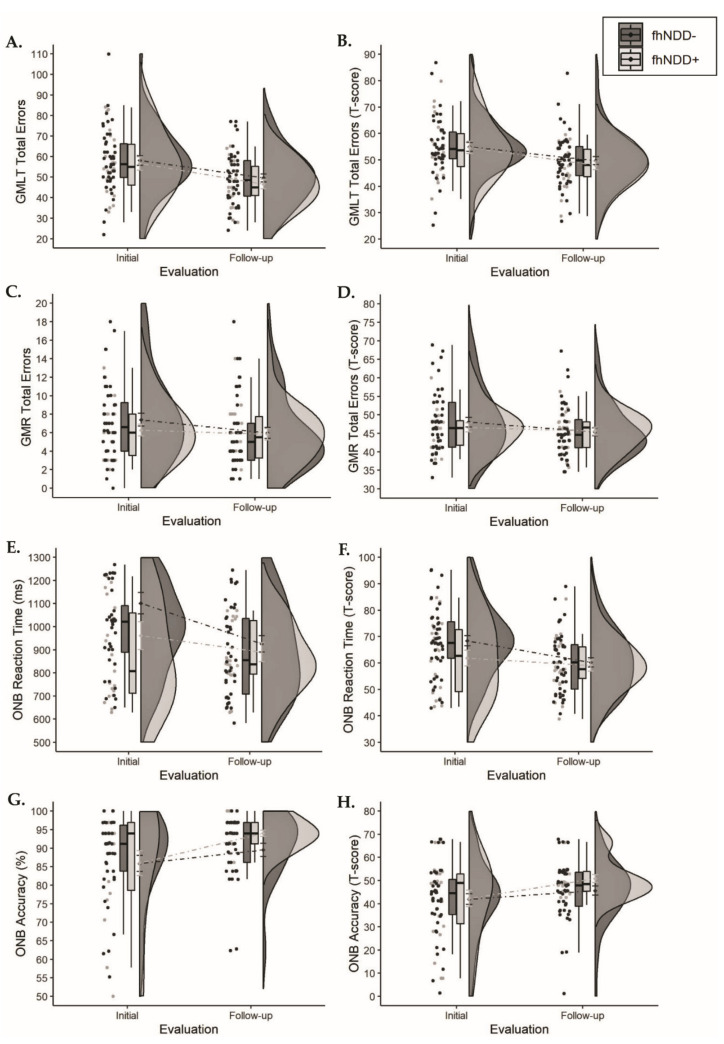
Group comparisons of cognitive performance between those with a family history of neurodegenerative disease (fhNDD+; light gray) and without a family history of neurodegenerative disease (fhNDD−; dark gray). (**A**) GMLT total errors. (**B**) GMLT total errors *t*-score. (**C**) GMR total errors. (**D**) GMR total errors *t*-scores. (**E**) ONB reaction time (ms). (**F**) ONB reaction time *t*-score. (**G**) ONB accuracy (%). (**H**) ONB accuracy *t*-score.

**Table 1 jcm-10-00528-t001:** Participant demographic information and injury characteristics.

	fhNDD− (*n* = 44)	fhNDD+ (*n* = 22)
Family history of Alzheimer’s disease N (%)	--	17 (77.3%)
Family history of Parkinson’s disease N (%)	--	6 (27.3%)
**Demographic Information**		
Age (years)	14.75 ± 1.71	15.00 ± 1.66
Body mass index (kg/m^2^)	23.84 ± 6.13	23.19 ± 5.05
Biological Sex N (%)		
Males	18 (40.9%)	9 (40.9%)
Females	26 (59.1%)	13 (59.1%)
Ethnicity, N (%)		
Caucasian	22 (50.0%)	16 (72.7%)
African American	18 (40.9%)	4 (18.2%)
Other/unknown	4 (9.1%)	2 (9.1%)
**Injury Characteristics**		
History of concussion, N (%)		
No history	32 (72.7%)	15 (68.2%)
One	12 (27.3%)	7 (31.8%)
Cause of injury, N (%)		
Sport or recreation	23 (52.3%)	14 (63.6%)
Motor vehicle accident	12 (27.3%)	4 (18.2%)
Other (fall, accident, etc.)	9 (20.4%)	4 (18.2%)
Time since injury (days)		
Initial evaluation	13.73 ± 7.85	14.41 ± 7.76
Follow-up evaluation	34.05 ± 12.59	35.14 ± 16.15

Note: data are reported as mean ± SD unless otherwise noted. Abbreviations: fhNDD, family history of neurodegeneration.

**Table 2 jcm-10-00528-t002:** Descriptive statistics for RPQ and Beck Youth Inventories—depression scale (BYI-D) scores at each evaluation.

		Evaluation		rmANOVA (η_p_^2^)		
Outcome Measures	Group	Initial	Follow-Up	Group	Time	Interaction
**RPQ (Total)**	fhNDD−	20.45 ± 2.02	11.20 ± 2.14	0.058	0.439 ^‡^	0.011
	fhNDD+	24.41 ± 2.14	12.55 ± 2.04			
Somatic	fhNDD−	10.80 ± 1.08	6.39 ± 1.19	0.066 *	0.398 ^‡^	0.003
	fhNDD+	13.68 ± 1.19	7.59 ± 1.33			
Emotional	fhNDD−	4.64 ± 0.58	2.55 ± 0.59	0.030	0.371 ^‡^	0.003
	fhNDD+	5.23 ± 0.67	2.73 ± 0.46			
Cognitive	fhNDD−	5.02 ± 0.53	2.23 ± 0.44	0.013	0.484 ^‡^	0.000
	fhNDD+	5.50 ± 0.66	2.23 ± 0.46			
**BYI-D**						
Raw Score	fhNDD−	6.20 ± 1.28	3.43 ± 0.79	0.019	0.141 ^‡^	0.000
	fhNDD+	6.36 ± 1.47	3.91 ± 0.84			
T-Score	fhNDD−	43.89 ± 8.98	41.57 ± 1.04	0.000	0.101 ^‡^	0.000
	fhNDD+	44.14 ± 1.65	41.59 ± 1.16			

Note: data are reported as group means ± SE and semi-partial η^2^ (η_p_^2^) measures of effect size for each rmANOVA. * Indicates statistical significance *p* ≤ 0.05, ^‡^ Indicates statistical significance *p* ≤ 0.01. Abbreviations: RPQ, Rivermead post-concussion symptoms questionnaire; BYI-D, Beck Youth Inventories—depression scale; fhNDD, family history of neurodegeneration.

**Table 3 jcm-10-00528-t003:** Descriptive statistics for heart rate variability (HRV) during the 5-min self-paced breathing assessment at each evaluation.

		Evaluation		rmANOVA (η_p_^2^)		
Outcome Measures	Group	Initial	Follow-Up	Group	Time	Interaction
**Time-domain**						
SDNN (ms)	fhNDD−	72.23 ± 4.16	69.58 ± 3.09	0.124 ^‡^	0.018	0.000
	fhNDD+	57.74 ± 4.27	54.15 ± 2.40			
RMSSD (ms)	fhNDD−	74.02 ± 4.41	71.37 ± 3.53	0.072 *	0.011	0.000
	fhNDD+	62.00 ± 5.61	58.87 ± 2.97			
**Frequency-domain**						
LF power (ln)	fhNDD−	7.00 ± 0.13	7.06 ± 0.12	0.137 ^‡^	0.002	0.010
	fhNDD+	6.55 ± 0.14	6.42 ± 0.16			
HF power (ln)	fhNDD−	7.69 ± 0.14	7.74 ± 0.12	0.123 ^‡^	0.011	0.026
	fhNDD+	7.27 ± 0.19	7.02 ± 0.17			

Note: data are reported as group means ± SE and semi-partial η2 (η_p_^2^) measures of effect size for each rmANOVA. * Indicates statistical significance *p* ≤ 0.05; ^‡^ Indicates statistical significance *p* ≤ 0.01. Abbreviations: HRV, heart rate variability; SDNN, standard deviation of NN intervals; RMSSD, root mean square of successive NN interval differences; LF, low-frequency; HF, high-frequency; ms, milliseconds; ln, natural logarithmic units; fhNDD, family history of neurodegeneration.

**Table 4 jcm-10-00528-t004:** Descriptive statistics for HRV during the 1-min resonant breathing assessments at each evaluation.

		Evaluation		rmANOVA (η_p_^2^)		
Outcome Measures	Group	Initial	Follow-Up	Group	Time	Interaction
**HRV_REST_**						
SDNN (ms)	fhNDD−	123.66 ± 5.56	117.38 ± 5.72	0.130 ^‡^	0.003	0.054
	fhNDD+	90.52 ± 5.08	100.36 ± 7.27			
RMSSD (ms)	fhNDD−	92.01 ± 5.20	91.08 ± 7.03	0.119 ^‡^	0.028	0.036
	fhNDD+	60.2 ± 3.62	75.20 ± 6.96			
**HRV_IHGC_**						
SDNN (ms)	fhNDD−	110.11 ± 5.46	114.76 ± 6.06	0.124 ^‡^	0.002	0.006
	fhNDD+	89.15 ± 5.80	87.93 ± 5.81			
RMSSD (ms)	fhNDD−	80.95 ± 5.18	93.79 ± 7.29	0.155 ^‡^	0.030	0.022
	fhNDD+	59.72 ± 3.36	60.63 ± 3.97			
**Change in HRV**						
∆SDNN (ms)	fhNDD−	−14.96 ± 2.75	−1.67 ± 3.28	0.004	0.008	0.147 ^‡^
	fhNDD+	−2.00 ± 2.26	−10.62 ± 4.86			
∆RMSSD (ms)	fhNDD−	−11.25 ± 3.09	4.27 ± 4.04	0.000	0.008	0.158 ^‡^
	fhNDD+	1.72 ± 3.03	−8.52 ± 5.51			

Note: data are reported as group means ± SE and semi-partial η2 (η_p_^2^) measures of effect size for each rmANOVA. ^‡^ Indicates statistical significance *p* ≤ 0.01. Abbreviations: HRV, heart rate variability; SDNN, standard deviation of NN intervals; RMSSD, root-mean-square of successive NN interval differences; HRV_REST,_ resting-state heart rate variability; HRV_IHGC_, heart rate variability during an isometric handgrip contraction; ∆, change between IHGC and rest; ms, milliseconds; fhNDD, family history of neurodegeneration.

**Table 5 jcm-10-00528-t005:** Descriptive statistics for CogState measures at each evaluation.

		Evaluation		rmANOVA (η_p_^2^)		
Outcome Measures	Group	Initial	Follow-Up	Group	Time	Interaction
**CogState Raw Scores**						
GMLT_ERRORS_	fhNDD−	58.10 ± 2.37	49.58 ± 1.96	0.007	0.327 ^‡^	0.003
	fhNDD+	56.63 ± 3.22	46.64 ± 2.30			
GMR_ERRORS_	fhNDD−	7.40 ± 0.69	5.97 ± 0.59	0.008	0.051	0.014
	fhNDD+	6.27 ± 0.64	5.82 ± 0.62			
ONB_RT_ (ms)	fhNDD−	1101.13 ± 46.07	925.64 ± 35.15	0.032	0.174 ^‡^	0.048
	fhNDD+	960.9 ± 60.31	889.4 ± 40.9			
ONB_ACC_ (%)	fhNDD−	85.85 ± 2.16	89.54 ± 1.78	0.012	0.154 ^‡^	0.027
	fhNDD+	85.96 ± 3.36	94.00 ± 0.86			
**CogState T-Scores**						
GMLT_ERRORS_	fhNDD−	54.91 ± 1.68	49.72 ± 1.54	0.003	0.258 ^‡^	0.006
	fhNDD+	54.66 ± 2.33	47.90 ± 1.69			
GMR_ERRORS_	fhNDD−	47.99 ± 1.29	45.37 ± 1.08	0.002	0.060 *	0.012
	fhNDD+	46.56 ± 1.21	45.55 ± 1.08			
ONB_RT_	fhNDD−	68.41 ± 1.95	60.19 ± 1.72	0.028	0.178 ^‡^	0.050
	fhNDD+	61.87 ± 2.99	59.08 ± 2.17			
ONB_ACC_	fhNDD−	41.98 ± 2.33	45.65 ± 1.92	0.012	0.160 ^‡^	0.029
	fhNDD+	42.31 ± 3.58	49.19 ± 1.80			

Note: data are reported as group means ± SE and semi-partial η2 (η_p_^2^) measures of effect size for each rmANOVA. * Indicates statistical significance *p* ≤ 0.05; ^‡^ Indicates statistical significance *p* ≤ 0.01 Abbreviations: GMLT_ERRORS_, Groton maze learning errors; GMR_ERRORS_, Groton maze delayed recall errors; ONB_RT_, One-Back task reaction time; ONB_ACC_, One-Back task accuracy; fhNDD, family history of neurodegeneration.

## Data Availability

The datasets generated for this study are available on request to the corresponding author.

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
