# Peer review of "The Influence of Family History of Neurodegenerative Disease on Adolescent Concussion Outcomes"

_jcm, 2021, doi:10.3390/jcm10030528_

Round 1
Reviewer 1 Report
This was a well designed and presented piece of research. I have a couple of suggestions.
- I appreciated being able to see a scatter plots in the group comparisons on HRV. I would recommend providing the same for the RPQ, BYI-D scores and CogState measures, as well, perhaps as a supplemental image.
- While, I understand the difficulties associated with being able to perform patient genotyping, I wonder if the authors could parse out any phenotypical variability between the different self-reported fhNDD categories (as in, family history of Alzheimer's disease vs PD). I realize that there will not be sufficient n to draw any reasonable conclusions but it may be interesting to evaluate, given you probably already have the data on hand.
Author Response
We appreciate and have addressed all the reviewer’s comments in the text and below. Per the editor’s recommendations, we have used track changes in the text, and all changes are referenced in the track changes (All markup) lines for each comment.
"Reviewer #1:This was a well designed and presented piece of research. I have a couple of suggestions. I appreciated being able to see a scatter plots in the group comparisons on HRV. I would recommend providing the same for the RPQ, BYI-D scores and CogState measures, as well, perhaps as a supplemental image."
We agree that visualizing the group comparisons with other measures is needed and have now added rain plots for the RPQ, BYI-D, and Cogstate measures in Figures 3 and 6.
"While, I understand the difficulties associated with being able to perform patient genotyping, I wonder if the authors could parse out any phenotypical variability between the different self-reported fhNDD categories (as in, family history of Alzheimer's disease vs PD). I realize that there will not be sufficient n to draw any reasonable conclusions but it may be interesting to evaluate, given you probably already have the data on hand."
We appreciate the reviewer’s comments and recognize that one genotype may more susceptible than the other. However, we believe that given insufficient sample size for analyses and that patient genotyping was not conducted, this is beyond the scope to the current study. We have added descriptive data for each category in Table 1 of the results. Additionally, we have tested the group differences between family history of Alzheimer's disease vs PD and did not observe any group differences in concussion outcomes. This is now briefly mention in the results in track changes lines 196-199. In addition, we now address this issue in the discussion. Please see track changes lines 347-348.
Reviewer 2 Report
The authors aim to answer an interesting and relevant medical question. Several factors have already been found to be associated with abnormal recovery from concussions in adolescents. These include sex and prior diagnosis of psychiatric or neurological conditions. In this study, the authors explore the link between a family history of neurodegenerative disease (fhNDD) and abnormal recovery from concussions in adolescents. The participant demographics were well laid out and variables controlled adequately. The study evaluates components of somatic, affective, and cognitive post-concussive symptoms as well as cardio-autonomic function. The data shows evidence on all of these measures, but not very extensive or in depth. The present study shows preliminary support for an effect of fhNDD on somatic symptoms and cardio-autonomic function after adolescent concussion. Their work did not generate support for an effect of fhNDD on mental health or cognition. The authors discuss that it is possible that their younger demonographics have more neural compensatory abilities compared to older individuals and hypothesize that may have been the cause for no correlation with cognitive ability. Are there additional clinical evaluations they could have performed, is it possible the task selected was not sensitive enough for the population? If these are analyses that cannot be added to the present work, then these options should be addressed in the discussion. However, this is minor and not the focus of the manuscript.
Overall, pertinent, interesting, and well-presented work. The work would benefit from additional testing options (as supplemental data or at least discussed) to evaluate the different parameters. Corroboration of 2+ tests would support the negative claims in the manuscript.
Authors do a good job at not overstating their conclusions and suggesting additional work is necessary to support widespread implementation of HRV for concussion patients.
Author Response
"Reviewer #2:The authors aim to answer an interesting and relevant medical question. Several factors have already been found to be associated with abnormal recovery from concussions in adolescents. These include sex and prior diagnosis of psychiatric or neurological conditions. In this study, the authors explore the link between a family history of neurodegenerative disease (fhNDD) and abnormal recovery from concussions in adolescents. The participant demographics were well laid out and variables controlled adequately. The study evaluates components of somatic, affective, and cognitive post-concussive symptoms as well as cardio-autonomic function. The data shows evidence on all of these measures, but not very extensive or in depth. The present study shows preliminary support for an effect of fhNDD on somatic symptoms and cardio-autonomic function after adolescent concussion. Their work did not generate support for an effect of fhNDD on mental health or cognition. The authors discuss that it is possible that their younger demographics have more neural compensatory abilities compared to older individuals and hypothesize that may have been the cause for no correlation with cognitive ability. Are there additional clinical evaluations they could have performed, is it possible the task selected was not sensitive enough for the population? If these are analyses that cannot be added to the present work, then these options should be addressed in the discussion. However, this is minor and not the focus of the manuscript."
We appreciate and have addressed all the reviewer’s comments in the text and below. Per the editor’s recommendations, we have used track changes in the text, and all changes are referenced in the track changes (All markup) lines for each comment.
We appreciate the reviewer’s comments and agree that the cognitive tasks selected may not be sufficient to capture group differences in cognitive function and future research is required to confirm our findings. This is now addressed in the discussion in track changes lines 334-337.
Further, we also agree that additional clinical evaluations in the context of the examined relationship, including those which measure vestibular and balance function, require further investigation. However, without adding considerable length to the manuscript and focus on these separate physiological pathways, we believe adding the insufficient data we have to be disadvantageous to the current study. That is, not all the participants included in the study have the VOMS and mBESS and we are under powered to analyze these measures. We now address the reviewer’s comments in the discussion track changes lines 356-359.